# Healthcare Access in the Aftermath: A Longitudinal Analysis of Disaster Impact on US Communities

**DOI:** 10.3390/ijerph22050733

**Published:** 2025-05-05

**Authors:** Kevin Chang, Jana A. Hirsch, Lauren Clay, Yvonne L. Michael

**Affiliations:** 1College of Medicine, Drexel University, 60 N. 36th Street, Philadelphia, PA 19104, USA; 2Urban Health Collaborative, Dornsife School of Public Health, Drexel University, Philadelphia, PA 19104, USA; jah474@drexel.edu; 3Department of Epidemiology and Biostatistics, Dornsife School of Public Health, Drexel University, Philadelphia, PA 19104, USA; ylm23@drexel.edu; 4Department of Emergency and Disaster Health Systems, University of Maryland Baltimore County, Baltimore, MD 21250, USA; lclay@umbc.edu

**Keywords:** environmental health, disasters, healthcare accessibility, longitudinal studies, United States, health infrastructure

## Abstract

Research on climate-related disasters and healthcare infrastructure has largely focused on short-term, localized impacts. This study examined the long-term association between climate-related disasters and healthcare facilities across 3108 contiguous United States counties from 2000 to 2014. Utilizing databases like the National Establishment Time Series and the Spatial Hazards and Events Losses Database, we classified county-level infrastructure changes (“never had”, “lost”, “gained”, and “always had”) and disaster severity (minor, moderate, severe), respectively. Autoregressive linear models were used to estimate the total number of moderate and severe disasters (2000–2013) associated with the change in the number of healthcare establishments in 2014, after adjusting for healthcare establishments, total population, and poverty in 2000. Results demonstrate that an increase in one moderate disaster was significantly associated with increased hospital infrastructure (Count, 0.14; 95% CI, 0.03–0.25), while severe disasters were significantly associated with a decrease (Count, −0.31; 95% CI, −0.47–−0.14). Similar but stronger associations were observed for ambulatory care (Moderate: Count, 2.52; 95% CI 0.91–4.12 and Severe: Count, −5.99; 95% CI, −8.53–−3.64, respectively). No significant associations were found among pharmacies. These findings highlight the varying impacts of climate-related disasters on healthcare accessibility. Future initiatives should prioritize strengthening existing infrastructure and enhance disaster recovery strategies.

## 1. Introduction

Climate change threatens our world’s ecosystem with more intense, frequent, and fatal weather-related disasters. In the United States (US) alone, fifteen or more billion-dollar weather-related events have occurred each year from 2015 to 2020 [1]. Approximately 24% of global deaths are due to modifiable environmental factors [2]. Climate-related disasters, such as extensive flooding or wind damage, can destroy crucial healthcare infrastructures in our communities, including pharmacies, hospitals, and ambulatory care centers [3]. As a result, healthcare may be disrupted, turning away patients and interrupting access to medication or routine appointments [4]. For example, Superstorm Sandy and the aftermath of Hurricane Katrina led to short-term health system closures, destruction of medical records, and a spike in medication demand, resulting in missed medical appointments and increased rates of uncontrolled hypertension [4]. Additionally, disasters obstruct continuity of care, worsening chronic health issues [5,6].

Post-disaster recovery efforts, which include the repairing or rebuilding of damaged infrastructure, require significant resources and collaboration between local and federal governments. However, communities that are of low socioeconomic status (SES) (e.g., low income) may experience greater challenges in their recovery. For example, communities have less insurance, savings, and personal resources to devote to their recovery [7]. Disaster-related losses in these communities have also been associated with worsening physical and mental health among those already at risk for poor health outcomes [7,8]. Worsening health-related conditions in low SES communities demonstrate the necessity for healthcare services that are both accessible and available post-disaster. However, an analysis examining the differences between Hurricane Katrina and Hurricane Sandy on healthcare provider availability demonstrated that access to care post-disaster remains inequitable [9]. After Katrina, researchers noted a county-level decline in primary care physicians, medical specialists, and surgeons—an effect not seen in affected counties after Sandy [9]. The median household income in counties at the time affected by Katrina ($45,800) was below the national average ($50,233), whereas the median income for counties affected by Sandy ($65,000) was well above ($56,516) [9]. These disparities underscore how county-level socioeconomic differences ultimately shape both acute disaster response and long-term recovery.

Research to date on post-disaster healthcare accessibility and availability has largely focused on single disasters and the short-term impacts of disrupted healthcare services on communities [10,11,12]. Additionally, past work has concentrated on the immediate effects of disasters on operations within specific healthcare facilities such as the staffing or availability of services in hospital centers [10,11,12]. Investigation of the long-term impact of climate-related disasters on diverse types of healthcare facilities at a larger, national scale has largely been unexplored [13]. Disasters significantly alter local healthcare infrastructure that can have lasting effects on human health. Shifting the focus beyond the immediate disaster response to the broader recovery period may reveal several crucial factors that influence health outcomes.

In our study, we address this gap by assessing the relationship between healthcare accessibility and disasters longitudinally throughout the US. To address this relationship, we focus on US counties, utilizing data from three primary datasets: healthcare infrastructure; disaster losses; and geographical polarization. We examine disaster losses over an extended period of time (2000 to 2014) and include data across various hazard types and levels of impact. In our study, we evaluate three large categories of healthcare infrastructure: (1) pharmacies; (2) hospital-based inpatient care; and (3) ambulatory care. Our research examines how disasters impact the way these healthcare institutions are distributed across the continental US over a 15-year time span, while controlling for confounding by race/ethnicity and SES.

## 2. Materials and Methods

### 2.1. Study Sample

This longitudinal cohort study linked data from three primary data sources to create an annual panel dataset of counties in the contiguous US from 2000 to 2014 totaling 46,620 county-years, including longitudinal data on disaster losses and healthcare infrastructure. The study had a total of 3108 continental nonwater US counties. To establish consistency over time, the 2000 US Census geographies were utilized for the demographic and socioeconomic measurements.

### 2.2. Measures

#### 2.2.1. Exposure Variables

Health and Healthcare Institutions. The presence of healthcare facilities was measured using 2000 to 2014 business data from the National Establishment Time Series (NETS) database, licensed from Walls & Associates (Denver, CO, USA) in January 2017 [14]. From the NETS, records were categorized as ambulatory care or as pharmacies using 8-digit Standard Industrial Classification (SIC) codes (Appendix A) according to methods published elsewhere [15]. Healthcare facilities were geocoded and aggregated to county for each year in which a business was open, focusing on 2000 and 2014. Using this information, we assessed counts of healthcare facilities per county and change over time. Categories of change in healthcare facilities were divided as follows: (1) Never (counties having no facilities at both time points (2000–2014)), (2) Lose (counties going from having at least one facility in 2000 to having none in 2014), (2) Gain (counties going from having no facilities in 2000 to having at least one in 2014), and (4) Always present (counties having at least one at both time points (2000–2014)). Due to the distribution of the NETS resource, ambulatory care values were measured using a threshold value of 10 facilities per county per year, rather than the initial threshold of one facility.

Disaster Impact. Disaster impact losses were measured using 2000 to 2014 data from the Spatial Hazards and Events Losses Database for the United States (SHELDUS^TM^), licensed from the Center for Emergency Management and Homeland Security at Arizona State University (Phoenix, AZ, USA), a national database of eighteen different hazard types at county level resolution for all 50 states [16]. SHELDUS^TM^ consolidates multiple disaster databases, including data from the National Climatic Data Center, US Geological Survey, and others. For every event with any measured loss, SHELDUS^TM^ provides: location (county), time (begin and end date), inflation-adjusted direct losses (property and crop damage, fatalities, injuries), and type of hazard (peril). SHELDUS^TM^ is a leading dataset relating to disasters and is used across dozens of studies [17,18,19]. We created a three-level typology: severe, moderate, and minor impact. For each county during 2000–2014, a severe disaster month was classified as greater than $50 property damage per capita and/or three or more fatalities, a moderate disaster month as between $10 and $50 of property damage per capita and/or two fatalities, and a minor disaster month occurs as fewer than $10 property damage per capita and/or has one fatality from disasters [20]. The disaster impact variable was created by totaling up the number of months in a year for each category for each county.

Geographical Polarization Data. Geographical polarization data was reported as a single metric value named the Index of Concentration at the Extremes (ICE). The ICE value represents the extent to which a county’s population are organized into the extremes of deprivation and privilege utilizing 2018 household income. The ICE value is a commonly used measure to assess disparities utilizing a geographical perspective [21]. Values range from −1 to +1, with a value of −1 indicating the most deprived county and a value of +1 indicating the most privileged county. For this project, we compared non-Hispanic white high-income groups versus person-of-color low-income groups. High-income counties were defined as populations that were part of the top quintile of household income and low-income counties were defined as populations that were part of the bottom quintile of household income [21].

#### 2.2.2. Covariates

Covariate data were assessed using data from the Longitudinal Tract Database provided by Brown University, including decennial census data from 2000 and 2010 and data from the National Historical Geographic Information Systems [22]. Longitudinal Tract Database data were produced at the census tract-level and were aggregated to the county-level. Using these data for each county, the annual community-level social and demographic characteristics were obtained: (1) the proportion of white residents, (2) poverty rate, measured as the percentage of individuals living below the federal poverty line, (3) owner-occupied homes, measured as the proportion of owner-occupied housing units (compared to renters), (4) proportion of college educated, measured as the proportion of residents aged 25 or older who have completed a bachelor’s degree, and (5) total county population. To avoid over-controlling, we identified a priori the most important confounding variables for inclusion in the analysis. Total population was included to adjust our estimates of change in the number of medical establishments relative to the baseline size of the county. For socioeconomic status, we selected poverty rate, based on prior disaster research and empirical evidence [7,8,23]. Since racial/ethnic composition of counties may reflect structural inequities in counties that relate to both vulnerability to climate-related disasters and the availability of healthcare resources, we also included the proportion of white residents [24]. By adjusting for total population, poverty rate, and non-Hispanic white residents, we aim to minimize residual confounding and better isolate the specific effects of disaster exposure on healthcare infrastructure over time and across geographic areas.

### 2.3. Statistical Analysis

We calculated descriptive statistics of healthcare infrastructure, demographic, socioeconomic, and geographical polarization characteristics for 2000, 2014 (healthcare infrastructure only), and 2018 (geographical polarization data only). Total counts of healthcare infrastructure were calculated for each county during the time period as well as the average proportion of such counties living below poverty, unemployed, college educated, or residing in owner-occupied housing units (Table 1). In addition, average ICE values of each county were calculated (Table 1). Because minor disasters encompassed a broad range of event hazards and were overrepresented in our dataset, they introduced considerable variability, making systematic analysis more challenging (Table 2). Given the study’s focus on more intense disasters with significant public health consequences, we limited our analyses to moderate and severe events to enhance clarity.

We fit multivariable linear models to estimate our primary outcome of interest: the change in the number of medical establishments in 2014 associated with a 1-unit increase in the number of years exposed to each type of moderate and severe disaster from 2000 to 2013 (Table 3). While the analysis does not follow a traditional longitudinal panel, this approach is characterized as a single-period change model with temporal ordering. This method, therefore, allows us to examine changes within a defined temporal framework. As mentioned above, poverty, total population, and non-Hispanic white residents were included in our adjusted models based on an a priori justification. Other variables such as race/ethnicity, homeownership, and education levels, were excluded to avoid overadjustment and multicollinearity. Additionally, to assess the robustness of our results, we conducted sensitivity analyses using change scores (the difference between the baseline (2000) and outcome (2014) values) as the dependent variable (Appendix A).

## 3. Results

### 3.1. Differences in Counties by Changes in Healthcare Facilities (2000–2014)

Counties that went from having at least one type of healthcare facility from 2000 to having none in 2014 had a higher proportion of residents living below poverty. Counties that experienced a loss of healthcare facilities, as well as those that never had such facilities at either time point, experienced consistently high levels of poverty. In contrast, counties that gained or always had a healthcare facility in their counties tended to have lower poverty rates. For example, counties that never had or lost pharmacies had poverty rates of 16.6% and 17.4%, respectively, while counties that gained or maintained facilities had rates of 15.3% and 14.0%, respectively (Table 1). A similar pattern was observed for both hospitals (16.1% and 17.6% vs. 15.0% and 13.8%) and ambulatory care (17.3% and 17.2% vs. 15.7% and 13.4%) (Table 1). These differences, while suggestive, are descriptive and not intended to imply causal associations. These differences may reflect broader socioeconomic differences between counties.

Counties that *lost* healthcare facilities had higher levels of segregation (lower ICE scores) compared to counties that consistently had healthcare facilities at both time points, across all categories of healthcare facilities (Table 1).

### 3.2. Changes in Disasters (2000–2014) by Changes in Healthcare Facilities (2000–2014)

Counties that lost their pharmacies (n = 66), hospitals (n = 86), and ambulatory care (n = 21), from 2000 to 2014, experienced more severe disasters. However, only counties that lost hospitals and ambulatory care facilities experienced more moderate disasters. These unadjusted differences are descriptive in nature and are not adjusted for confounding; they are not intended to suggest causality.

### 3.3. Change in Total Number of Healthcare Establishments in 2014 with Total Number of Moderate and Severe Disasters

While an increase in one moderate disaster was associated with a statistically significant increase in hospital infrastructure over the time period (Count, 0.14; 95% CI, 0.03–0.25), we observed a significant decrease in hospital infrastructure (Count, −0.31; 95% CI, −0.47–−0.14) associated with severe disaster. We observed a similar pattern of association for ambulatory care (Moderate: Count, 2.52; 95% CI, 0.91–4.12 and Severe: Count, −5.799; 95% CI, −8.35–−3.64, respectively). We observed no significant association between moderate or severe disasters in relation to pharmacies (Table 3).

The sensitivity analyses using change scores confirmed the strength, direction, and significance of the associations observed with our primary model, reinforcing the robustness of our findings. These results are presented in Appendix A.

## 4. Discussion

This study describes changes in healthcare availability associated with climate-related disasters across the contiguous US between 2000 and 2014. Few prior studies have evaluated post-disaster changes in different types of healthcare facilities at the national level.

Our research found that over the 15-year period, counties that always lacked access to healthcare services or lost their facilities were more likely to have high poverty levels (Table 1). Additionally, counties that lost their healthcare facilities experienced higher racial segregation, indicated by lower ICE values (Table 1). Especially in rural areas, the closure of healthcare facilities often worsens economic instability [25,26,27]. As care is delayed, chronic conditions like cardiovascular disease and cognitive decline become more difficult to manage [5,28,29,30]. On the other hand, counties that always had or gained healthcare services over the 15-year period were more likely to experience lower poverty levels (Table 1). Counties that always had their healthcare facilities exhibited less racial segregation, indicated by higher ICE values (Table 1). These communities often have economic and political influence that enable them to secure resources/policies that maintain their healthcare infrastructure [26,31,32,33].

Nevertheless, when disaster recovery strategies are effectively coordinated between the federal and local level, they yield significant benefits. After Hurricane Irma in 2017, the state of Florida and the Federal Emergency Management Agency (FEMA) instituted the Coordinated Place-Based Recovery Support (CPBRS) initiative to support local leadership in the most affected counties with tailored state and federal recovery resources. The CPBRS model demonstrates a locally executed, state-managed, and federally supported framework for disaster response. The success of this effort demonstrated the possibility of establishing cross-agency disaster recovery protocols that are efficient and equitable, particularly assisting historically underserved communities. For example, its goals closely aligned with local needs and priorities such as infrastructure improvements or water system upgrades in the most impacted counties [34].

The association between disasters and changes in availability of healthcare facilities also varies by the intensity of disaster and the category of healthcare facility. Counties that experienced a loss of healthcare facilities across all types from 2000 to 2014 also experienced a greater increase in severe climate-related disasters (Table 2). Even after adjusting for potential confounders, a significant decrease in healthcare facilities—particularly hospitals and ambulatory care—was observed in these counties following each severe climate-related disaster. As the severity of Hurricane Katrina demonstrated, many facilities were either not rebuilt or their services reduced, leading to disruptions to necessary medical care [35].

Rebuilding facilities after severe climate-related disasters is further complicated by financial constraints and coordination among governmental agencies [36]. In low-income communities and communities of color, recovery efforts are often systematically excluded from access to rebuilding resources [37]. Thus, among communities that are already afflicted with pre-existing healthcare accessibility challenges, the destructive impact of disasters may widen existing disparities [38,39,40]. A study among Hurricane Ike survivors found that those with a lower annual income and a high school degree or equivalent were more likely to experience depression post-climate-related disaster compared to wealthier, more educated survivors [7,8]. In another case, the aftermath of Hurricane Katrina led to disruptions to access to care for those with chronic diseases. Medical records were lost due to water damage, and one of the most frequently cited challenges was medication procurement, specifically its availability, due to the spike in demand post-disaster [41].

In contrast to our findings regarding severe disasters, moderate disasters were associated with a significant increase in hospitals and ambulatory care throughout the time period. This idea fits the notion that disasters can, in some cases, lead to (possibly unequal) redevelopment [42,43,44]. Moderate disasters may provide governing agencies with an opportunity to learn and develop policies that prioritize investment in and the construction of resilient infrastructure, proactively addressing the risks of future, more severe climate-related events. For instance, traditional “build-back-better” efforts and disaster response protocols often leave communities vulnerable to future hazards due to the tendency to rebuild to prior, inadequate engineering standards [45,46,47]. Novel initiatives, like Alternative Project Delivery Methods, a process involving earlier collaboration between designers and contractors, can leverage stakeholders in the community to develop more robust and resilient infrastructure reconstruction [48].

Finally, we observed the absence of an association between the presence of pharmacies and moderate or severe disasters over time. We presume that the absence is largely due to the already existing low number of pharmacies within parts of the US, termed “pharmacy deserts” [49,50]. Especially as online pharmacies become more prevalent, communities become increasingly reliant on delivery infrastructure rather than the physical presence of pharmacies. The frequent closures over the past decade driven by the growing role of pharmacy benefit managers (PBMs), have also reduced access, leading to poorer health outcomes and increased costs from hospitalizations and emergency room visits [51,52]. Therefore, the shifting dynamics of pharmacy accessibility demonstrate nuances in how healthcare systems are impacted by disasters, influencing health outcomes and community recovery.

### Strengths and Limitations

This study utilized a longitudinal dataset to analyze healthcare infrastructure in association with climate-related disasters. Linking four national datasets consisting of demographic, social stratification, healthcare infrastructure, and climate-related disaster events between 2000 and 2014, the findings provide relevant information to inform the development of resilient communities with robust healthcare infrastructure.

There are some limitations within this study. First, data licensing agreements limited the healthcare infrastructure data available after 2014. However, the relations we examined between disaster and access to healthcare infrastructure may be conservative given the increasing intensifying effect of climate-related disasters [53,54,55]. Second, our operationalization of demographic, healthcare infrastructure, or climate-related disaster variables may have resulted in misclassification. However, we used datasets and variables that have been validated in previous research [21,56,57,58,59]. Third, there may be unmeasured confounders that we did not account for with our autoregressive model such as the effects of systemic racism along with preexisting geographic vulnerability. For example, historical redlining is associated with communities having poor health outcomes lacking access to healthcare facilities from practices like the systemic closures of hospitals or the inaccessibility of pharmacies [60,61,62]. Such communities are often located in areas prone to flooding or other environmental hazards [63,64]. As a result, delayed or even absent recovery efforts can further discourage trust in the medical system and potentially delay necessary medical care [65,66]. Controlling for or exploring these effects may involve incorporating redlining maps and additional geospatial data and is a focus for future research. Our modeling approach assumes that change in healthcare infrastructure began after the baseline year. However, if counties were already on a declining or growing trajectory before 2000, and those trends continued, adjusting for baseline levels may introduce horse-racing bias—a spurious association between disaster exposure and change due to prior, unmeasured trends [67]. Our sensitivity analysis produced similar results, reducing concern related to this bias. We acknowledge this potential limitation and interpret findings with caution. Fourth, the racial composition may introduce a limitation, as a wide majority of counties (85.9%) are predominantly white, as shown in Table 1. However, the distribution reflects the overall demographic composition of the U.S., with findings that are broadly representative of national trends. While regional differences in disaster recovery may exist, future research could explore such nuances in more diverse populations. A more detailed analysis, however, is beyond the scope of this study.

## 5. Conclusions and Further Research

Building on the limitations noted above, we have identified several promising directions for future studies. While our study sought to test associations between climate-related disasters and healthcare infrastructure, future research should focus on testing the potential underlying mechanisms driving the trends we observed. Factors such as policy response, funding availability, and community resilience measures are promising areas of future study. Additionally, based on prior research, our study examined three key types of healthcare infrastructures: pharmacies, hospitals, and ambulatory care. However, future work could expand our analysis to other facilities, including community health centers, urgent care facilities, or long-term care institutions. Finally, additional socioeconomic and demographic variables, such as healthcare workforce availability or rural–urban disparities, could be evaluated as modifiers of the association between disasters and healthcare infrastructure.

The study examines the temporal change in the availability of various healthcare institutions across the contiguous U.S. counties associated with climate-related disasters. Combining national data from 2000 to 2014, our findings describe the differential impact of climate-related disasters on disruptions to healthcare facilities. Hospital and ambulatory care are less accessible in counties that experienced a greater number of severe climate related disasters. Given the US’s diversity in geography, population, and policy, it is important to examine the factors that influence how communities respond and recover from disaster. Developing innovative strategies, such as an effective coordination between local, state, and federal resources, to enhance resilience within healthcare infrastructure can protect public health and reduce healthcare disparities during increasingly severe disasters.

## Figures and Tables

**Table 1 ijerph-22-00733-t001:** Demographic and socioeconomic characteristics (2000) and climate-related disasters (2000) across categories of change (2000–2014) in healthcare facilities for 3108 continental nonwater US counties.

	**No. (%)**								
		**Pharmacies**				**Hospitals**			
**2000 Characteristic ^a^**	**All counties** **(n = 3108)**	**Never ^e^** **(n = 116)**	**Lose** **(n = 66)**	**Gain** **(n = 28)**	**Always** **(n = 2898)**	**Never** **(n = 307)**	**Lose** **(n = 86)**	**Gain** **(n = 150)**	**Always** **(n = 2565)**
Race/ethnicity ^b^									
Predominantly NH white	2671 (85.9)	96 (82.8)	51 (77.3)	24 (85.7)	2500 (86.3)	267 (87.0)	63 (73.3)	129 (86.0)	2212 (86.2)
Predominantly NH black	45 (1.4)	2 (1.7)	2 (3.0)	0	41 (1.4)	5 (1.6)	5 (5.8)	3 (2.0)	32 (1.2)
Predominantly Hispanic/Latino	33 (1.1)	4 (3.4)	2 (3.0)	0	27 (0.9)	4 (1.3)	2 (2.3)	5 (2.7)	23 (0.9)
Other or Racially/ethnically mixed	359 (11.6)	14 (12.1)	11 (16.7)	4 (14.3)	330 (11.4)	37 (10.1)	16 (18.6)	14 (9.3)	298 (11.6)
Living below poverty, mean (SD), %	14.2 (6.5)	16.6 (8.9)	17.4 (8.5)	15.3 (8.8)	14.0 (6.3)	16.1 (7.4)	17.6 (7.5)	15.0 (7.6)	13.8 (6.2)
Unemployed, mean (SD), %	6.2 (3.4)	6.0 (6.6)	5.9 (4.2)	6.2 (4.2)	6.2 (3.1)	5.7 (3.8)	6.9 (3.4)	6.2 (3.2)	6.2 (3.3)
College Educated, mean (SD), %	16.5 (7.8)	16.4 (6.0)	15.6 (6.3)	15.6 (7.8)	16.5 (7.9)	14.0 (5.6)	12.3 (3.9)	13.8 (7.7)	17.1 (8.1)
Ownership-Occupied Housing Units, mean (SD), %	74.1 (7.6)	73.2 (10.4)	74.8 (5.8)	76.2 (6.0)	74.1 (7.5)	77.0 (6.8)	76.0 (8.4)	77.9 (6.1)	73.5 (7.5)
Total Population, mean (SD)	89,956.1 (293,542.8)	3058.1 (2892.6)	6659.0 (4560.4)	8853.4 (7253.6)	96,115.0 (303,068.1)	7433.3 (6428.5)	10,635.9 (6740.9)	17,713.2 (12,661.53)	106,717.2 (320,604.4)
Index of Concentration at the Extremes (ICE)									
ICEwnhinc ^c^, mean (SD)	0.10 (0.13)	0.09 (0.18)	0.07 (0.09)	0.11 (0.19)	0.11 (0.13)	0.10 (0.07)	0.02 (0.11)	0.09 (0.11)	0.11 (0.10)
Climate-related Disaster (total number) ^d^									
Minor Disaster	5355	42	55	19	5239	267	72	203	4813
Moderate Disaster	603	36	14	5	548	66	15	19	503
Severe Disaster	305	18	11	4	272	39	6	11	249
			**No. (%)**						
			**Ambulatory Care ^e^**						
**2000 Characteristic**			**Never** **(n = 495)**		**Lost** **(n = 21)**		**Gain** **(n = 201)**	**Always** **(n = 2391)**	
Race/ethnicity									
Predominantly NH white			403 (81.4)		15 (71.4)		172 (85.6)	2081 (87.0)	
Predominantly NH black			14 (2.8)		2 (9.5)		6 (3.0)	23 (1.0)	
Predominantly Hispanic/Latino			12 (2.4)		1 (4.8)		1 (0.5)	19 (0.8)	
Other or Racially/ethnically mixed			66 (13.3)		3 (14.3)		22 (10.9)	268 (11.2)	
Living below poverty, mean (SD), %			17.3 (7.7)		17.2 (8.7)		15.7 (7.1)	13.4 (6.0)	
Unemployed, mean (SD), %			5.9 (4.4)		7.8 (4.5)		6.3 (3.6)	6.2 (3.1)	
College Educated, mean (SD), %			13.3 (4.7)		13.0 (3.4)		13.0 (4.6)	17.5 (8.3)	
Ownership-Occupied Housing Units, mean (SD), %			76.2 (7.0)		77.5 (5.3)		78.0 (4.8)	73.3 (7.5)	
Total Population, mean (SD)			5585.2 (3751.8)		9285.9 (3415.9)		10,466.9 (4657.25)	114,813.9 (330,653.4)	
Index of Concentration at the Extremes (ICE)									
ICEwnhinc, mean (SD)			0.07 (0.09)		0.04 (0.08)		0.08 (0.13)	0.12 (0.09)	
Climate-related Disaster (total number)									
Minor Disasters			354		30		246	4725	
Moderate Disasters			110		4		50	439	
Severe Disasters			66		6		22	211	

^a^ Data are from the 2000 US Census report unless otherwise indicated. ^b^ Racial/ethnic composition was assessed by predominant (>60%) racial/ethnic group. Places with no predominant group were classified as racially/ethnically mixed areas. ^c^ Index of Concentration at the Extremes (ICE) for non-Hispanic white high-income vs. person of color low-income 2018 value. High-income is defined as counties that were part of the top quintile of household income and low-income is defined as counties that were part of bottom quintile of household income. The ICE variable ranges from −1 (least privileged) to +1 (most privileged). ^d^ Data are from the SHELDUS^TM^ national database unless otherwise indicated. ^e^ Due to the distribution of the NETS resource, ambulatory care values were defined as: (1) Never (counties having less than 10 facilities at both time points (2000–2014)), (2) Lost (counties that had at least 10 facilities in 2000 to less than 10 in 2014), (3) Gain (counties that had less than 10 facilities in 2000 to at least 10 in 2014), and (4) Always (counties always having at least 10 facilities at both time points (2000–2014)).

**Table 2 ijerph-22-00733-t002:** Change in climate-related disaster occurrences (2000–2014) across categories of change in healthcare facilities (2000–2014) for 3108 continental nonwater US counties.

	**No. (%)**				
		**Pharmacies**			
**Change in Characteristic (2000–2014)**	**All counties** **(n = 3108)**	**Never** **(n = 116)**	**Lose** **(n = 66)**	**Gain** **(n = 28)**	**Always** **(n = 2898)**
Climate-related Disaster, mean (SD),%:					
Minor Disaster	7.5 (221.1)	−13.8 (105.4)	19.7 (129.2)	−21.4 (87.6)	−7.7 (227.0)
Moderate Disaster	−5.8 (7.5)	18.1 (61.3)	10.6 (43.4)	17.9 (39.0)	5.1 (58.7)
Severe Disaster	−2.1 (42.5)	1.7 (47.5)	3.0 (58.1)	−3.6 (57.6)	2.1 (41.7)
		**Hospitals**			
**Change in Characteristic (2000–2014)**		**Never (n = 307)**	**Lose (n = 86)**	**Gain** **(n = 150)**	**Always** **(n = 2565)**
Climate-related Disaster, mean (SD),%:					
Minor Disaster		9.4 (146.0)	33.7 (192.0)	2.5 (208.8)	5.3 (230.0)
Moderate Disaster		−7.5 (60.3)	4.6 (59.2)	−0.7 (53.7)	−6.2 (58.4)
Severe Disaster		−2.3 (49.5)	3.5 (38.9)	−1.3 (30.6)	−2.3 (42.3)
		**Ambulatory Care**			
**Change in Characteristic (2000–2014)**		**Never** **(n = 495)**	**Lose** **(n = 21)**	**Gain** **(n = 201)**	**Always** **(n = 2391)**
Climate-related Disaster, mean (SD),%:					
Minor Disaster		13.1 (138.0)	−47.6 (143.6	−0.50 (165.4)	7.4 (239.0)
Moderate Disaster		−5.9 (62.6)	4.6 (63.2)	−6.0 (75.3)	−5.8 (55.8)
Severe Disaster		−1.2 (49.7)	3.5 (53.9)	0 (40.0)	−2.2 (41.0)

**Table 3 ijerph-22-00733-t003:** Adjusted autoregressive models measuring change in total number of healthcare establishments in 2014 with total number of moderate and severe climate-related disasters for 3108 nonwater US counties.

	Pharmacies			Hospitals			Ambulatory Care		
	Count (*β*)	SE	95% CI	Count (*β*)	SE	95% CI	Count (*β*)	SE	95% CI
Climate-related Disasters									
Moderate Disaster	0.09	0.08	(−0.06, 0.24)	0.14 *	0.06	(0.03, 0.25)	2.52 **	0.82	(0.91, 4.12)
Severe Disaster	0.10	0.11	(−0.12, 0.32)	−0.31 ***	0.08	(−0.47, −0.14)	−5.99 ***	1.20	(−8.35, −3.64)
Model Fit	R^2^ = 0.97			R^2^ = 0.97			R^2^ = 0.99		
N=	3108								

* *p* < 0.05; ** *p* < 0.01; *** *p* < 0.001. Counts were adjusted for poverty rates in 2000, the total number of healthcare establishments in 2000, total population in 2014, and non-Hispanic white residents.

## Data Availability

The original contributions presented in this study are included in the article/Appendix A. Further inquiries can be directed to the corresponding author(s).

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
