# Peer review of "Healthcare Access in the Aftermath: A Longitudinal Analysis of Disaster Impact on US Communities"

_ijerph, 2025, doi:10.3390/ijerph22050733_

Round 1
Reviewer 1 Report
Comments and Suggestions for Authors
Please check the attachment

Author Response
Comments 1:
- While the study establishes significant associations, it does not fully explore the mechanisms behind these trends. For instance, why do moderate disasters lead to infrastructure gains, while severe disasters result in losses? Future research could delve into factors such as policy responses, funding availability, and community resilience measures.
Response 1: We value the reviewers’ comment regarding exploring further the mechanisms behind our observed trends. Our study primarily aimed to establish significant associations, and while factors like policy responses, funding availability, and community resilience measures likely influence the observed trends, a data-based exploration of such mechanisms goes beyond the scope of this study. In this revision, we not only included some possible mechanisms as described in other research in our discussion of our findings (see lines 270-276, 300-320) but also created an additional section labeled “Future Research” that addresses some potential avenues for further exploration:
“Building on the limitations noted above, several promising directions for future studies emerge. While our study primarily sought to test associations between climate-related disasters and healthcare infrastructure, future research should focus on exploring the underlying mechanisms driving the trends we observed. Factors such as policy response, funding availability, and community resilience measures are promising areas of future study. Additionally, our study examined three key types of healthcare infrastructures: pharmacies, hospitals, and ambulatory care. Future work could expand our analysis to include other facilities, including community health centers, urgent care facilities, or long-term care institutions. Finally, future research may incorporate additional socioeconomic and demographic variables that went beyond what we studied to evaluate effect modification, such as healthcare workforce availability or rural-urban disparities”. (see p. 10, lines 336-348).
Comments 2:
- The analysis would benefit from a more detailed breakdown of healthcare infrastructure beyond hospitals, ambulatory care, and pharmacies. Including metrics on community health centers, urgent care facilities, or long-term care institutions could provide a more comprehensive picture.
Response 2: We agree with the reviewers’ comment regarding a more detailed breakdown of healthcare infrastructure beyond the three categories we examined. Our goal of this study was to provide a broader picture and focus on key access to healthcare infrastructure elements in the community impacted by disasters in prior research: pharmacies, hospitals, and ambulatory care (see lines 39-52). While a more granular analysis of community health centers, urgent care facilities, or long-term care institutions would provide additional insights, incorporating such elements goes beyond the intended scope of the study. However, we do indeed recognize the importance of this perspective and included this point in our newly added “Future Research” section (see p. 10, lines 336-348).
Comments 3:
- While the conclusion emphasizes the need for enhanced disaster recovery strategies, specific policy suggestions could further strengthen the study’s impact. For example, discussing potential interventions such as federal funding allocation models or resilience-building initiatives at the local level would add practical value.
Response 3: We have added more information to the discussion regarding a specific policy intervention that adds practical value:
“Nevertheless, when disaster recovery strategies are effectively coordinated between the federal and local level, they yield significant benefits. After Hurricane Irma in 2017, the state of Florida and the Federal Emergency Management Agency (FEMA) instituted the Coordinated Place-Based Recovery Support (CPBRS) initiative that supported local leadership in the most affected counties with tailored state and federal recovery resources. Such initiative proved beneficial as they were closely aligned with local needs and priorities such as infrastructure improvements or water system upgrades.” (see p. 8, lines, 255-261)
Reference:
Federal Emergency Management Agency. Locally executed, state-managed, federally supported recovery. Published March 12, 2021. Accessed March 31, 2025. https://www.fema.gov/case-study/locally-executed-state-managed-federally-supported-recovery
Comments 4:
- Although the study adjusts for healthcare establishments, total population, and poverty in 2000, other socioeconomic and demographic variables (e.g., insurance coverage, healthcare workforce availability, or rural-urban disparities) could influence the observed trends. Addressing these factors in future research would enhance the robustness of the conclusions.
Response 4: We appreciate the reviewers’ insightful comment regarding examining other socioeconomic and demographic variables that could influence the observed trends. After a review of current literature, we identified a priori important confounding variables total population and poverty in 2000. Adjusting for additional factors such as insurance coverage, healthcare workforce availability, or rural-urban disparities could lead to over-control of our association of interest, given their correlation with the covariates already included to control for confounding. However, we acknowledge their potential impact and value for consideration in future research. For additional clarity, we revised the methods portion of our manuscript under the Covariates section:
“Covariate data was assessed using data from the Longitudinal Tract Database provided by Brown University, including decennial census data from 2000 and 2010 and data from the National Historical Geographic Information Systems [22]. Longitudinal Tract Database data were produced at the census tract-level and were aggregated to the county-level. Using these data for each county, the annual community-level social and demographic characteristics were obtained: (1) proportion of white residents, (2) poverty rate, measured as the percentage of individuals living below the federal poverty line, (3) owner-occupied homes, measured as the proportion of owner-occupied housing units (compared to renters), (4) proportion college educated, measured as the proportion of residents aged 25 or older who have completed a bachelor’s degree, and (5) total county population. To avoid over-controlling, we identified a priori the most important confounding variables for inclusion in the analysis. Total population was included to adjust our estimates of change in the number of medical establishments relative to the baseline size of the county. For socioeconomic status, we selected poverty rate, based on prior disaster research and empirical evidence [7 8 23]. By adjusting for total population and poverty rate, we aim to minimize residual confounding and better isolate the specific effects of disaster exposure on healthcare infrastructure over time and across geographic areas.” (see p. 4 lines 141-158)
Comments 5:
This paper provides a well-executed and highly relevant analysis of how climate-related disasters influence healthcare accessibility in the US. By identifying disparities in infrastructure resilience following disasters, the study lays a strong foundation for future research and policy interventions. While some areas could be expanded, particularly regarding mechanisms and policy implications, the findings remain highly valuable for disaster preparedness and healthcare infrastructure planning.
Response 5: We appreciate the reviewer’s belief in the importance of this research.
Reviewer 2 Report
Comments and Suggestions for Authors
Short Summary
The Authors assess the long-run relationship between healthcare accessibility and disasters in the US. Specifically, they examine how disasters impact the way pharmacies, hospitals, and ambulatories are distributed across US over a 15-year time span at county level (3108 counties examined) and control for confounding by race/ethnicity and SES. The Authors compute and comments descriptive statistics of the phenomena under inquiry, then estimate through multivariate linear models the change in the number of medical establishments in 2014 associated with a change in the number of years exposed to each type of moderate and severe disaster from 2000 to 2013. They find that counties that always lacked access to healthcare services or lost their facilities were more likely to have high poverty levels and racial segregation. On the other hand, counties that gained or maintained their facilities were predominantly non-Hispanic White and had higher education levels. The association between disasters and changes in availability of healthcare facilities also varies by the intensity of disaster and the category of healthcare facility.
Broad comments
The Authors illustrate research on the relationship between healthcare accessibility and disasters in the US. While potentially of interest to the readership of the journal, in my opinion the manuscript lacks of clarity. Specifically, results are not clearly presented and methods are not adequately described. Also, conclusions need to be further elaborated.
a. Concerning methods, descriptive statistics should be presented and commented properly.
1. In Table 1 it can be noticed how a wide majority of counties is predominantly white (85.9% of the sample). This evidence should be commented, as the "disproportion" might affect the interpretation of results (see comments about "Discussion"). Also, it is not clear why the Authors reported data collected for 2000 instead of a mean for the whole period 2000-2014 (or, at least, also data for 2014). Finally, all socioeconomic variables considered in the analysis and measured for each kind of healthcare infrastructure and "pattern" should be briefly commented or deleted.
2. In Table 1 it is not clear why Column "All counties" only illustrates data about pharmacies, while none is said about hospitals and ambulatories.
3. In the methodology the Authors are invited to further elaborate on the multiple linear models adopted in the empirical analysis. Specifically, they are invited to write the functional specification of the models (the regression equation), the rationale of their choice, and the meaning of the variables included in the models. Also they are invited to specify the dimensions of the longitudinal dataset and the software used to elaborate the data.
b. Concerning results, many statements are not confirmed in the data illustrated in Table 1 and 2. Specifically:
1. Authors state that "Across all types of healthcare categories, pharmacies experienced the least change (n=94)." I could not find this data in Tables 1 and 2. Authors are invited to specify where to find this data in Tables 1 and 2.
2. Authors claim that "Counties that lost hospitals had the highest proportion of individuals living below poverty (17.6%). We observed similar levels of poverty among counties that never had a type of healthcare facility at both time points. (Table 1)". In fact, it seems that the proportion of individuals living below poverty does not significantly change when considering other healthcare facilities and "patterns".
3. Authors claim that "Counties that lost their pharmacies (n=66), hospitals (n=86), and ambulatory care 198 (n=21), 2000 to 2014, experienced more moderate and severe disasters". This statement is only partially correct, as counties that lost their pharmacies seem to have experienced less moderate disasters than counties in other categories.
4. The output of the autoregressive models is not properly presented in table 3. Specifically, Authors are invited to write the functional specification of the models in Section Methodology (see comment 1c), and to report in Section Results at least: the dependent variable of each model, the beta coefficients, their standard error, the R^2 of the models, and the number of observations.
c. In the Discussion, Authors are invited to connect their considerations to the data reported in tables 1 and 2. Specifically:
1. Authors claim that "over the 15-year period, counties that always lacked access to healthcare services or lost their facilities were more likely to have high poverty levels and racial segregation". I cannot see a connection between lack of healthcare services and racial segregation
2. Authors claim that "counties that gained or maintained their facilities were predominantly non-Hispanic White and had higher education levels". I cannot see a connection between the two data, especially considering that in data illustrated in table 1 and 2 the non-Hispanic White are disproportionately represented. Authors are invited to explain how to read the data in Tables 1 and 2 to find evidence in support of their thesis.
3. Other claims that should be supported by the data illustrated in Tables 1 and 2 are the following:
i. "among all categories of facilities, pharmacies experienced the most stability during the period";
ii. "Counties that experienced a loss of healthcare facilities across all types from 2000-2014 also experienced the greatest increase in moderate and severe climate-related disasters".
iii. "we observed the absence of an association between the presence of pharmacies and moderate or severe disasters over time".
Short summary
Concluding remarks need to be further elaborated. Specifically:
- what are the policy implications of this research?
- what suggestions for further research?
Author Response
Broad comments
a. Concerning methods, descriptive statistics should be presented and commented properly.
- In Table 1 it can be noticed how a wide majority of counties is predominantly white (85.9% of the sample). This evidence should be commented, as the "disproportion" might affect the interpretation of results (see comments about "Discussion"). Also, it is not clear why the Authors reported data collected for 2000 instead of a mean for the whole period 2000-2014 (or, at least, also data for 2014). Finally, all socioeconomic variables considered in the analysis and measured for each kind of healthcare infrastructure and "pattern" should be briefly commented or deleted.
Response 1: We appreciate the reviewers’ comment regarding the racial composition of the sample (85.9%). However, the distribution reflects the overall demographic composition of the U.S., with findings that are broadly representative of national trends. We have added this point in our “Strengths and Limitations” section:
“Additionally, the racial composition may introduce a limitation as a wide majority of counties (85.9%) are predominantly white, as shown in Table 1. However, the distribution reflects the overall demographic composition of the U.S., with findings that are broadly representative of national trends.” (see p. 10, lines 329-334)
Regarding the use of 2000 data instead of a mean for the whole period, the study utilized the baseline values for covariates (2000 Census) because these factors are known predictors of the number of medical establishments at both baseline and follow-up, thus including it in the model helps isolate the effect of the exposure on the change in the outcome.
We also appreciated the reviewers’ comment regarding the inclusion of all socioeconomic variables in the analysis and measured for each kind of healthcare infrastructure and “pattern.” We identified, a priori, the most important confounding variables to avoid overcontrolling for factors that might impact our analysis. The variables we selected were identified after drawing on previous disaster research and empirical evidence that aligned with the study’s objectives. Inclusion of other socioeconomic factors may lead to over-control of the effect of interest due to correlation with our identified confounding factors.
- In Table 1 it is not clear why Column "All counties" only illustrates data about pharmacies, while none is said about hospitals and ambulatories.
Response 2: We greatly appreciate the reviewers’ attention to detail. The data for hospitals and ambulatory care facilities is presented later in the table. The table was structured in this way to improve readability and organization. However, we have adjusted the formatting of Table 1 and reoriented the table horizontally for ease of readability. (see p. 5-6)
- In the methodology the Authors are invited to further elaborate on the multiple linear models adopted in the empirical analysis. Specifically, they are invited to write the functional specification of the models (the regression equation), the rationale of their choice, and the meaning of the variables included in the models. Also they are invited to specify the dimensions of the longitudinal dataset and the software used to elaborate the data.
Response 3: We acknowledge the reviewer’s suggestion. We did not include the regression equation in the manuscript, as it was not consistent with the journal’s expected readership. However, we are more than happy to provide the functional specification here:
Facility= β_0+β_1 (Moderate Disasters)+ β_2 (Severe Disasters)+β_3 (Poverty)+β_4 (Healthcare Establisment)+ β_5 (Population)+ϵ
Where Facility represents the number of healthcare establishment (pharmacy, hospital, or ambulatory care) in 2014, Moderate Disasters is the sum of 2000-2013 disasters classified as moderate, Severe Disasters is the sum of 2000-2013 classified as severe, Poverty represents the poverty rates in 2000, Healthcare Establishment represents the total number of healthcare establishments (pharmacy, hospital, or ambulatory care) in 2000, and Population is the total population in 2014. The error term (ϵ) captures unobserved variable.
We utilized a multivariable linear model because our outcome of healthcare facility in 2014, or dependent, variable is continuous, allowing us to estimate the effects of multiple predictors while controlling for confounding. The variables that were selected were guided by prior literature to avoid overcontrolling. Additionally, inclusion of the baseline level of healthcare establishments allows for the multivariable model to estimate the marginal change in facility counts, accounting for pre-existing differences across counties. This approach helps to isolate the specific effect of an exposure on the outcome.
Regarding the dimensions of the longitudinal dataset, the dataset contains 3,108 continental nonwater United States counties over 15 years (2000 -2014) with 14 variables per county per year. The software used to elaborate the data was RStudio Version 2022.2.3.492.
b. Concerning results, many statements are not confirmed in the data illustrated in Table 1 and 2. Specifically:
1. Authors state that "Across all types of healthcare categories, pharmacies experienced the least change (n=94)." I could not find this data in Tables 1 and 2. Authors are invited to specify where to find this data in Tables 1 and 2.
Response 4: We thank the reviewers for their comment. The value n=94 for pharmacies was obtained by summing up the values in two different columns in Table 1. Specifically: Column Lose (n=66) + Column Gain (n=28) = 94.
This value reflects the total number of counties experiencing change in the number of healthcare establishments (pharmacies, hospitals, ambulatory care). However, upon reflection we have decided to remove this point from the manuscript. The previous interpretation relied on raw counts of counties, which did not account for the baseline characteristics of the county or population size. We recognize that this comparison may not offer meaningful conclusions for readers and to avoid overinterpretation of raw numbers, we have removed this statement.
- Authors claim that "Counties that lost hospitals had the highest proportion of individuals living below poverty (17.6%). We observed similar levels of poverty among counties that never had a type of healthcare facility at both time points. (Table 1)". In fact, it seems that the proportion of individuals living below poverty does not significantly change when considering other healthcare facilities and "patterns".
Response 5: We appreciate the reviewers’ comment regarding the poverty levels in counties that lost their healthcare facilities across all categories. Upon further review of Table 1, we acknowledge that the pattern of poverty levels is similar across healthcare facilities. Higher poverty levels were associated with counties that never had or lost their facilities, whereas lower levels were associated with counties that gained or always had facilities. To address this, we have revised our statement to avoid any confusion. The updated text is as follows:
“Counties that experienced a loss of healthcare facilities, as well as those that never had such facilities at either time point, experienced consistently high levels of poverty. In contrast, counties that gained or always had a healthcare facility in their counties tended to have lower poverty rates. For example, counties that never had or lost pharmacies had poverty rates of 16.6% and 17.4%, respectively, while counties that gained or maintained facilities had rates of 15.3% and 14.0%, respectively (Table 1). A similar pattern was observed for both hospitals (16.1% and 17.6% vs. 15.0% and 13.8%) and ambulatory care (17.3% and 17.2% vs. 15.7% and 13.4%) (Table 1).” (see p. 7, lines 206-215)
This revision clarifies that counties experienced similar trends with respect to the change of their healthcare facilities over time. We appreciate the reviewers’ comment, it helped to improve the interpretation of our results.
- Authors claim that "Counties that lost their pharmacies (n=66), hospitals (n=86), and ambulatory care 198 (n=21), 2000 to 2014, experienced more moderate and severe disasters". This statement is only partially correct, as counties that lost their pharmacies seem to have experienced less moderate disasters than counties in other categories.
Response 6: We acknowledge and appreciate the reviewers’ careful examination of our statement. It is correct that counties that lost their pharmacies seem to have experienced fewer moderate disasters than counties in other categories. However, our analysis also supports that the counties that lost healthcare facilities overall experienced more severe disasters. To ensure accuracy, we have revised and updated our statement:
“Counties that lost their pharmacies (n=66), hospitals (n=86), and ambulatory care (n=21), from 2000 to 2014, experienced more severe disasters. However, only counties that lost hospitals and ambulatory care facilities experienced more moderate disasters.” (see p. 7, lines 220-222)
The revision provides additional clarification regarding the distinction. We thank the reviewer for their feedback to ensure the accuracy of our results.
- The output of the autoregressive models is not properly presented in table 3. Specifically, Authors are invited to write the functional specification of the models in Section Methodology (see comment 1c), and to report in Section Results at least: the dependent variable of each model, the beta coefficients, their standard error, the R^2 of the models, and the number of observations.
Response 7: We thank the reviewers’ request to provide additional specification of our models in Section Methodology as well as clearer presentation of our models in Table 3. In response we have clarified the functional specification of the models in our previous response to comment 1c. Furthermore, we did not include the standard error or the R2 of the models, as the presentation of such values does not align with the conventions typically expected by the readership of the journal. The 95% confidence interval in an autoregressive model output additionally provides a more intuitive and informative way to represent uncertainty around the estimated coefficients rather than the standard error. The confidence intervals provide a range in which the true population parameter is likely to fall whereas the standard error only indicates the variability of the point estimate. However, for further clarification and accuracy we have reiterated the formula and provided such values below for each of the healthcare facility categories:
〖Facility〗_Pharmacy= β_0+β_1 (Moderate Disasters)+ β_2 (Severe Disasters)+β_3 (Poverty)+β_4 (Healthcare Establisment)+ β_5 (Population)+ϵ
Dependent Variable: Number of Pharmacies in 2014
Beta Coefficients and Standard Error:
- Intercept: -4.179 (SE=0.5509)
- Moderate Disasters: 0.08633 (SE=0.07627)
- Severe Disaster: -0.1073 (SE=0.1119)
- Poverty (2000): 6.525 (SE=3.297)
- Healthcare Establishment (Pharmacies): 1.120 (SE=0.01490)
- Population (2014): 0.00007109 (SE=0.000002094)
R2 of models:
- Multiple R2: 0.9749
- Adjusted R2: 0.9747
Dependent Variable: Number of Hospitals in 2014
Beta Coefficients and Standard Error:
- Intercept: -2.160 (SE=0.4090)
- Moderate Disasters: 0.1386 (SE=0.05657)
- Severe Disaster: -0.2826 (SE=0.08312)
- Poverty (2000): 3.679 (SE=2.449)
- Healthcare Establishment (Hospitals): 1.346 (SE=0.02519)
- Population (2014): 0.00005757 (SE=0.000001694)
R2 of models:
- Multiple R2: 0.9671
- Adjusted R2: 0.9671
Dependent Variable: Number of Ambulatory Care Facilities in 2014
Beta Coefficients and Standard Error:
- Intercept: 4.550 (SE=5.912)
- Moderate Disasters: 2.491 (SE=0.8174)
- Severe Disaster: -5.775 (SE=1.199)
- Poverty (2000): -0.01098 (SE=0.3525)
- Healthcare Establishment (Ambulatory Care): 1.210 (SE=0.01143)
- Population (2014): 0.0009748 (SE=0.00002449)
R2 of models:
- Multiple R2: 0.988
- Adjusted R2: 0.9879
Number of Observations:
- 3,108 Continental Nonwater United States Counties
c. In the Discussion, Authors are invited to connect their considerations to the data reported in tables 1 and 2. Specifically:
1. Authors claim that "over the 15-year period, counties that always lacked access to healthcare services or lost their facilities were more likely to have high poverty levels and racial segregation". I cannot see a connection between lack of healthcare services and racial segregation
Response 8: We appreciate the reviewers’ comment. The connection between the lack of healthcare services and racial segregation can be found in Table 1. We have revised the statement in the manuscript for additional clarity:
“Our research found that over the 15-year period, counties that always lacked access to healthcare services or lost their facilities were more likely to have high poverty levels. (Table 1) Additionally, counties that lost their healthcare facilities experienced higher racial segregation, indicated by lower ICE values. (Table 1)” (See p. 8, lines 242-245)
- Authors claim that "counties that gained or maintained their facilities were predominantly non-Hispanic White and had higher education levels". I cannot see a connection between the two data, especially considering that in data illustrated in table 1 and 2 the non-Hispanic White are disproportionately represented. Authors are invited to explain how to read the data in Tables 1 and 2 to find evidence in support of their thesis.
Response 9: We appreciate with the reviewers’ comment and the opportunity to provide additional information. To improve clarity and maintain consistency with our earlier statement (see comment above), we have decided to remove the reference to education levels. Instead, we have revised the statement to focus on poverty and ICE values, which aligns clearer with the patterns observed in Table 1.
“On the other hand, counties that always had or gained healthcare services over the 15-year period were more likely to experience lower poverty levels. (Table 1) Counties that always had their healthcare facilities exhibited less racial segregation, indicated by higher ICE values. (Table 1)” (See p. 8, lines 248-251)
The revision ensures that our claims are supported by the data. Let us know if further clarification is needed.
- Other claims that should be supported by the data illustrated in Tables 1 and 2 are the following
i. "among all categories of facilities, pharmacies experienced the most stability during the period";
Response 10: We appreciate the reviewers’ comment. In Table 1, pharmacies had the greatest number of counties that belonged to the “Never” and “Always” columns. This indicates that from 2000 to 2014, a large majority of pharmacies were the most stable (experienced the least change) within those counties. However, as we mentioned in our prior response (see Response 4), we have decided to remove this point.
ii. "Counties that experienced a loss of healthcare facilities across all types from 2000-2014 also experienced the greatest increase in moderate and severe climate-related disasters".
Response 11: We appreciate the reviewers’ comment and agree with the need for clarification for this statement. Upon further review, we found that the reporting of moderate climate-related disasters was incorrect. To ensure accuracy and provide improved clarity, we removed the reference to moderate disasters and have revised the statement:
“Counties that experienced a loss of healthcare facilities across all types from 2000-2014 also experienced a greater increase in severe climate-related disasters. (Table 2)” (See p. 8-9, lines 262-264)
iii. "we observed the absence of an association between the presence of pharmacies and moderate or severe disasters over time".
Response 12: We appreciate the reviewers’ comment. The absence of an association between the presence of pharmacies and moderate or severe disasters over time can be found in Table 3. Our analysis demonstrated no significant associations between moderate or severe disasters in relation to pharmacies. We are happy to provide additional clarification if needed.
Short summary
Concluding remarks need to be further elaborated. Specifically:
- what are the policy implications of this research?
Response 13: We acknowledge the reviewers’ comment. We have added more information to the discussion regarding a specific policy intervention that adds practical value and provides a real-world recommendation that policymakers could potentially draw on.
“Nevertheless, when disaster recovery strategies are effectively coordinated between the federal and local level, they yield significant benefits. After Hurricane Irma in 2017, the state of Florida and the Federal Emergency Management Agency (FEMA) instituted the Coordinated Place-Based Recovery Support (CPBRS) initiative that supported local leadership in the most affected counties with tailored state and federal recovery resources. Such initiative proved beneficial as they were closely aligned with local needs and priorities such as infrastructure improvements or water system upgrades.” (see p. 8, lines, 255-261)
Reference:
Federal Emergency Management Agency. Locally executed, state-managed, federally supported recovery. Published March 12, 2021. Accessed March 31, 2025. https://www.fema.gov/case-study/locally-executed-state-managed-federally-supported-recovery
- what suggestions for further research?
Response 14: We appreciate the reviewers’ suggestions for further research. We have included a new section titled “Future Research.” Please refer to p. 10, lines 337-348, for these details. We are happy to provide additional clarification if needed.
Reviewer 3 Report
Comments and Suggestions for Authors
Dear authors,
Considering climate change, this is a valuable study. Some suggestions are:
A structured abstract should be written.
The data is old, which is why new data is not used.
It is suggested that data mining techniques be used to understand the study.
Keywords should be based on the mesh and also corrected.
In the introduction, can you explain the reason for choosing and focusing on three factors?
Methods
The type of study and the method of analysis should be specified precisely.
On what basis is the selection of criteria based?
Criteria, input, and output should be specified correctly as titles.
The discussion should be reviewed based on the findings.
The method of acknowledgement and thanks should be specified.
Author Response
Considering climate change, this is a valuable study. Some suggestions are:
A structured abstract should be written.
Response 1: We appreciate the reviewers’ suggestion. As per the journal’s requirements, the abstract should be a single paragraph and should follow the style of a structured abstract without headings. We have ensured that our abstract aligned with the journal’s requirements.
It is suggested that data mining techniques be used to understand the study.
Response 2: We appreciate the reviewers’ suggestion. Certainly, there is an increased use of data mining techniques, which has led to significant innovation in the field of statistics and machine learning. However, this falls outside the scope of our current study but remains an important focus for future research.
Keywords should be based on the mesh and also corrected.
Response 3: We agree with the reviewers’ suggestion. We have revised our Keywords section to be based on MESH. We revised the section as follows: Environmental Health; Disasters; Health Care Accessibility; Longitudinal Studies; United States; Health Infrastructure. (see p. 1, lines 32-33)
In the introduction, can you explain the reason for choosing and focusing on three factors?
Response 4: We appreciate the reviewers’ comment. We chose and focused on pharmacies, hospitals, and ambulatory care facilities based on prior literature review, indicating that these three types of healthcare facilities offer key essential services that directly influence human health outcomes (see lines 39-52). Additionally, when large-scale disasters impact communities, they inherently impact healthcare infrastructure. These facilities are critical for not only providing emergency medical care but also maintaining continuity of care. Disruptions to such medical establishments can lead to adverse public health consequences as a result.
Methods
The type of study and the method of analysis should be specified precisely.
Response 5: We appreciate the reviewers’ comment. For ease of clarity, we have included the following statement in the methods section.
“This longitudinal cohort study linked data from three primary data sources to create an annual panel dataset…” (See p. 2, lines 88)
Additionally, the method of analysis is outlined on p. 4, lines 159-177, where we describe our statistical analysis methods as well as our multivariable linear models.
On what basis is the selection of criteria based?
Response 6: We appreciate the reviewers’ comment on our selection of criteria. We sought to avoid overcontrolling during our multivariable linear regression analysis. Thus, we identified a priori the most important confounding variables that drew on previous disaster research and empirical evidence. By adjusting for such variables, we aimed to minimize residual confounding (see p. 4, lines 142-158).
Criteria, input, and output should be specified correctly as titles.
Response 7: We agree with the reviewers’ suggestion on specifying the criteria, input, and output as titles. To ensure clarity in our titles and ease of flow to readers we have revised and adjusted the following titles in the manuscript:
Study Sample Criteria
2.2 Measures
2.2.1. Exposure Variables
Health and Healthcare Institutions. …
Disaster Impact. …
Geographical Polarization Data. …
2.2.2. Covariates. …
(See p. 2-4, lines 86-141)
Furthermore, in the Statistical Analysis section, we revised the following statement to provide clarity about our outcome measure of interest (output):
“We fit multivariable linear models to estimate our primary outcome of interest: the change in the number of medical establishments in 2014 associated with a 1-unit increase in the number of years exposed to each type of moderate and severe disaster from 2000 to 2013 (Table 3).” (see p. 4, lines 171-174)
The discussion should be reviewed based on the findings.
Response 8: We acknowledge the reviewers’ comment on the discussion being reviewed based on the findings. We have integrated our findings into the Discussion section by highlighting significant relationships and providing possible mechanisms such as the varying impacts of moderate vs. severe climate-related disasters on different types of healthcare facilities. More information can be found on p. 8-10, lines 237-348, for reference.
The method of acknowledgement and thanks should be specified.
Response 9: We thank the reviewers for their suggestion on specifying the method of acknowledgement. As part of the journal submission process, we have already included the acknowledgments section.
Round 2
Reviewer 2 Report
Comments and Suggestions for Authors
I would like to thank the Authors for having considered my comments in the elaboration of the revised manuscript. However, I have noticed as many replies to my comments signal potential shortcomings in the underlying research that need to be fixed.
- The "disproportion" of predominantly white counties (or other "disproportions" as the huge number of counties that "always" had health care facilities) can be fixed by referring to standardized populations, and/or by using econometric techniques of matching. This passage is necessary to avoid that statements concerning the impact of some socioeconomic characteristics may reflect biases due to the use of a severely umbalanced sample.
- To improve clarity, Authors are invited to represent data concerning "All counties" (see 2) separately using histograms or other graphical tools.
- The statistical analysis has serious flaws and needs to be structurally revised. The model presented by the Authors in "Response 3" is a multivariate (not autoregressive, not multivariable) regression model, that uses cross-country data (cumulated, delayed, etc.) to explain the level of facilities in 2014. In the end, when performing the regression analysis, the dataset is not longitudinal, but sectional. Furthermore, it is rather naive to explain the level of a variable using the 14-lag delayed level of the same variable. Statistical literature may suggest, instead, to estimate a beta convergence (changes in the variable explained by the initial levels), or an autoregressive model using a wider set of lags of the dependent variable as regressors.
- The revised statement illustrated in "Response 5" does not provide a satisfactory solution to my comment, as differences in poverty rates does not seem significative and may reflect differences in the size and other characteristics of the populations considered.
- The output of the regression analysis must be presented according to the widely aknowledged statistical standards, otherwise it is impossible for the reader to understand the underlying regression analysis. The R^2 close to one signal a spurious regression, and this is probably due to the inclusion of a lagged value of the dependent variable among the regressors. Based on these premises, Authors are invited to structurally revise their empirical analysis and illustrate results according to the widely aknowledged statistical standards.
- Conclusions must be further elaborated. They should include: a short summary of the paper (few lines are enough), research limitations and suggestions for further research (Authors are invited to merge the subsection "Future research" in the conclusions), policy and managerial implications (Authors are invited to further elaborate on the proposition illustrated in Response 13).
Reviewer 3 Report
Comments and Suggestions for Authors
Dear authors
Thank you for your thorough and thoughtful revisions. Your responses to the reviewers’ comments are clear and appropriate. Your chosen topic is interesting and relevant, and your manuscript demonstrates significant merit.
Based on your revisions and the subject matter's importance, I believe your paper is suitable for publication. I look forward to seeing your valuable contribution published and making an impact in the field.
Best regards,
Author Response
Dear Reviewer:
Thank you so much for your kind words and thoughtful consideration of our manuscript. We greatly appreciate your time, thoughtful review, and constructive suggestions, which have helped strengthen our manuscript. We’re thankful for your support and glad to hear that the revised manuscript meets your expectations.